# Prognostic Significance of Lymphocyte Infiltration and a Stromal Immunostaining of a Bladder Cancer Associated Diagnostic Panel in Urothelial Carcinoma

**DOI:** 10.3390/diagnostics10010014

**Published:** 2019-12-28

**Authors:** Hideki Furuya, Owen T.M. Chan, Kanani Hokutan, Yutaro Tsukikawa, Keanu Chee, Landon Kozai, Keith S. Chan, Yunfeng Dai, Regan S. Wong, Charles J. Rosser

**Affiliations:** 1Clinical and Translational Research Program, University of Hawaii Cancer Center, Honolulu, HI 96813, USA; hideki.furuya@cshs.org (H.F.); khokutan@cc.hawaii.edu (K.H.); ytsukikawa@hawaii.edu (Y.T.); kchee@cc.hawaii.edu (K.C.); 2Department of Molecular Biosciences and Bioengineering, University of Hawaii at Manoa, Honolulu, HI 96822, USA; 3Department of Surgery, Cedars Sinai Medical Center, Los Angeles, CA 90048, USA; rwongega@ucla.edu; 4John A. Burn School of Medicine, University of Hawaii at Manoa, Honolulu, HI 96813, USA; lkozai@hawaii.edu; 5Department of Pathology, Cedars Sinai Medical Center, Los Angeles, CA 90048, USA; keith.chan@cshs.org; 6Department of Biostatistics, University of Florida, Gainesville, FL 32611, USA; ydai@ufl.edu

**Keywords:** bladder cancer, stroma, lymphocyte, prognosis

## Abstract

We set out to expand on our previous work in which we reported the epithelial expression pattern of a urine-based bladder cancer-associated diagnostic panel (A1AT, ANG, APOE, CA9, IL8, MMP9, MMP10, PAI1, SDC1, and VEGFA). Since many of the analytes in the bladder cancer-associated diagnostic signature were chemokines, cytokines, or secreted proteins, we set out to report the stromal staining pattern of the diagnostic signature as well as CD3^+^ (T-cell) cell and CD68^+^ (macrophage) cell staining in human bladder tumors as a snapshot of the tumor immune landscape. Immunohistochemical staining was performed on 213 tumor specimens and 74 benign controls. Images were digitally captured and quantitated using Aperio (Vista, CA). The expression patterns were correlated with tumor grade, tumor stage, and outcome measures. We noted a positive correlation of seven of the 10 proteins (excluding A1AT and IL8 which had a negative association and VEGFA had no association) in bladder cancer. The overexpression of MMP10 was associated with higher grade disease, while overexpression of MMP10, PAI1, SDC1 and ANG were associated with high stage bladder cancer and CA9 was associated with low stage bladder cancer. Increased tumor infiltration of CD68^+^ cells were associated with higher stage disease. Overall survival was significantly reduced in bladder cancer patients’ whose tumors expressed eight or more of the 10 proteins that comprise the bladder cancer diagnostic panel. These findings confirm that the chemokines, cytokines, and secreted proteins in a urine-based diagnostic panel are atypically expressed, not only in the epithelial component of bladder tumors, but also in the stromal component of bladder tumors and portends a worse overall survival. Thus, when assessing immunohistochemical staining, it is important to report staining patterns within the stroma as well as the entire stroma itself.

## 1. Introduction

Bladder cancer, the fifth most common malignancy in the US, will be diagnosed in approximately 80,470 patients (with a preponderance of these patients being males over the age of 60 years) and will result in approximately 17,670 deaths in 2019 [1]. The majority of newly diagnosed bladder cancer (70%) are non-muscle invasive bladder cancer (NMIBC), which is disease confined to the mucosa and submucosal tissues. In patients with NMIBC, intravesical Bacillus Calmette-Guérin (BCG) or chemotherapy are considered as the treatment of choice resulting in improved progression-free intervals after initial tumor resection [2]. Unfortunately, despite these therapies, patients with NMIBC are at high risk for the recurrence of tumors (70%), leading to high prevalence of bladder cancer in the US, second only to colorectal cancer [3,4]. In addition to this high rate of tumor recurrence, 30% of patients with NMIBC will progress to muscle invasive bladder cancer (MIBC), which is associated with a reduced overall survival (5-year survival < 50%) compared to NMIBC (5-year survival > 90%), whilst another 50% of these NMIBC patients will undergo removal of their bladders in an attempt to control their disease [5]. In the management of NMIBC, no new drug has been FDA approved in over 15 years [6]. Radiation with concomitant chemotherapy may be the primary treatment option for MIBC outside the US. However, in the US, radical cystectomy is the preferred treatment modality for these tumors, but unfortunately up to 50% of patients experience disease relapse and eventual death [7,8,9]. Improvement of these outcomes will require greater understanding of bladder cancer tumorigenesis and progression.

Identifying molecular variations in bladder tumorigenesis may lead to improved tools to diagnose as well as treat patients with bladder cancer. Recent studies in which primary bladder tumors were profiled demonstrated that, on the molecular level, bladder cancer can be further divided into three intrinsic subtypes referred to as luminal, basal and p53-like [10]. These molecular subtypes recapitulate the expression pattern of markers corresponding to intermediate/superficial layers (luminal subtype) and basal layers (basal subtype) of normal urothelium. Despite these recent advances, no clinical practice changing paradigms have resulted. Biopsy and subsequent genomic profiling of progressing and/or metastatic bladder lesions is not routinely performed. This is unfortunate as this is what patients die from, not the primary disease. Previously, we reported on a urine-based bladder cancer associated diagnostic panel comprised of 10 targets: Alpha-1 Antitrypsin (A1AT), Apolipoprotein E (APOE), angiogenin (ANG), carbonic anhydrase 9 (CA9), Interleukin 8 (IL8), matrix metallopeptidase 9 (MMP9), matrix metallopeptidase 10 (MMP10), Plasminogen activator inhibitor-1 (PAI1), Syndecan-1 (SDC1), and Vascular endothelial growth factor A (VEGFA) [11,12,13,14,15,16,17,18].

The composite biomarkers of the diagnostic panel have a variety of attributed functions. These include angiogenesis, degradation of extracellular matrix, serine protein inhibitor, catalyzation of the reversible hydration of carbon dioxide, lipid metabolism, and cellular signaling, while the two predominant functions include angiogenesis (IL8, VEGFA and ANG) and degradation of extracellular matrix (MMP9 and MMP10). In addition to IL8, VEGFA and ANG, MMP9, MMP10 and PAI1 have been linked to angiogenesis [19,20,21]. Angiogenesis, the development of new blood vessels from existing blood vessels, is necessary for normal tissue growth and the development of organs. A balance of angiogenic factors favoring angiogenesis and angiogenic factors opposing angiogenesis tightly govern this process [22,23,24]. In cancers, often times, this balance may favor angiogenic factors favoring angiogenesis, which then allows for sustained abnormal growth of tissues [25]. Besides MMP9 and MMP10 degrading extracellular matrix, recent reports have noted the ability of ANG and PAI1 to degrade the extracellular matrix [26,27,28]. The degradation of the extracellular matrix allows tumor cells to become more motile, which could lend itself to the development of metastatic disease. Previously, we reported the immunostaining patterns of the 10 targets within a bladder cancer -associated diagnostic panel in the epithelial component of human bladder tumors and noted a strong association between the expression of these targets and the presence of malignancy [28]. We hypothesized that the expression of the above diagnostic panel may correlate tissue T-cell and macrophage levels since we clearly know that T-cells (CD3^+^) play a major role in bladder cancer as well as macrophages (CD68^+^). Herein, we report on the presence of these 10 biomarkers in the stroma of human bladder cancer and to determine the clinical significance of the signature.

## 2. Materials and Methods

### 2.1. Patients and Clinicopathologic Information

The retrospective study with a waiver of consent was approved by Western Institutional Review Board (IRB #Rosser 2014-1). The cohort consisted of 287 patients; 213 with bladder cancer and 74 controls (no cancer; 47 autopsies for non- bladder cancer cause of death and 27 bladder neck specimens from radical prostatectomy specimens) seen at our institute from 1/1/2005 to 1/12/2010. No patient was treated with neoadjuvant chemotherapy prior to the collection of tissue. Diagnostic paraffin embedded tissues blocked from these patients were collected. Medical records were queried for age, race, sex, treatment and cancer related death. Furthermore, histology (urothelial carcinoma only), tumor grade (2002 WHO classification) and tumor stage (2002 TNM classification) were collected from medical records and confirmed by reevaluation of the original pathology slides. Clinical characteristics of the 287 subjects comprising the study cohort are illustrated in Table 1.

### 2.2. Immunohistochemistry

Immunohistochemical staining was performed as previously reported [28]. Briefly, paraffin blocks were cut into 5 μm sections and placed on a Superfrost Plus slides. Sections were deparaffinized then subjected to antigen retrieval using citric acid buffer (pH 6.0, 95 °C for 20 min). Next, slides were immersed in hydrogen peroxide (1%) with methanol. After 20 min blocking with 5% horse serum, slides were incubated overnight at 4 °C with the following primary antibodies: anti-PAI1 (#HPA050039; rabbit monoclonal, dilution 1:100) from Sigma Aldrich (St. Louis, MO,USA); anti-VEGFA (A-20) (#sc-152; rabbit polyclonal, dilution 1:500) and anti-ANG (#sc-74528; mouse monoclonal, dilution 1:10) from Santa Cruz Biotechnology, Inc. (Dallas, TX, USA); anti-SDC1 (B-A38) (#ab34164, mouse monoclonal, dilution 1:400); anti-MMP9 (EP1254) (#ab76003, rabbit monoclonal, dilution 1:200); and anti-MMP10 (#ab38930, rabbit polyclonal, dilution 1:2000) from Abcam (Cambridge, MA, USA); anti-CA9 (#23300002, rabbit polyclonal, dilution 1:1000) and anti-A1AT (#NBP1-90309, rabbit polyclonal, dilution 1:2500) from Novus Biologicals (Littleton, CO, USA); anti-APOE (D12) (#M068-3, mouse monoclonal, dilution 1:200) from MBL Co. (Nagoya, Japan), anti-IL8 (#AHC0881, rabbit polyclonal, dilution 1:200) from Life Technologies, Inc. (Grand Island, NY, USA), anti-CD3 (#ab16669, rabbit monoclonal, dilution 1:100); and anti-CD68 (#ab955, mouse monoclonal, dilution 1:200) from Abcam. Next, slides were incubated with 2 µg/mL of biotinylated anti-mouse or anti-rabbit IgG secondary antibodies (Vector Laboratories, Burlingame, CA, USA) for 30 min at room temperature. Lastly, sections were treated with Standard Ultra-Sensitive ABC Peroxidase Staining kit (Pierce/Thermo Fisher Scientific, San Jose, CA, USA) and 3,3′-diaminobenzidine (DAB; Vector Laboratories), counterstained with hematoxylin.

The following served as positive controls: human lung (PAI1 and MMP9), human liver (VEGFA, ANG, CA9 and A1AT), human tonsil (SDC1, MMP10 and APOE), human stomach (IL8), and human lymph node (CD3^+^ and CD68^+^), while omitting the primary antibody acted as the negative control (Appendix A).

### 2.3. Image Analysis

The slides were scanned using the Aperio Scanscope Cs (Aperio Technologies, Vista, CA, USA) as high-resolution images (20× objective). Images were visualized using Image Scope (Aperio, Vista, CA, USA). The immunoreactivity staining patterns within the stroma were noted. A prescribed algorithm developed by Aperio was used to assess staining intensity of the tissue and percent of cells/tissue staining for each target within the stromal only. Notably, we have previously used an algorithm to report the staining within the epithelial component of the tumor [28]. For statistical purposes, the data for each target was divided into quartiles: 1st quartile having the lowest staining intensity (0–10%) and the fourth quartile having the highest staining intensity (>50%). The data and slides were then reviewed and corroborated independently by an experienced pathologist (OTMC).

### 2.4. Statistical Analysis

The relationship between the immunoexpression of the 12 targets within the stroma and clinicopathological features were analyzed by Chi-square or Fisher’s test, and all tests were two-tailed. Kaplan-Meier curves were constructed using the log-rank test to estimate and compare disease-specific survival based on the immunoexpression of stroma and epithelial. Multivariate analysis using Cox proportional hazards models for overall survival was performed to evaluate the influences of age, sex, race, tumor grade, tumor stage and immunostaining on disease-specific survival. All statistical tests were two-sided with significance set at a *p* value of <0.05. SAS V9.4 (Cary, NC, USA) was used to perform statistical analyses.

## 3. Results

### 3.1. Demographics of the Patients and Tumor Characteristics

The age of the bladder cancer subjects ranged from 30 to 94 years (mean ± SD, 71.8 ± 11.9), while all of the control subjects were less than 65 years. Seventy-two percent of bladder cancer subjects were male and 76.5% of the cancer patients were Caucasian, while 78% of control subjects were male. Seventy patients (33%) with bladder cancer had a history of bladder cancer (recurrence). Twelve percent of bladder cancer subjects had tumors > 5 cm, while 23% had tumors < 2 cm. All tumors were confirmed to be urothelial carcinoma. The tumors were classified as either low-grade (26; 12.2%) or high-grade (176; 82.6%) as well as non-muscle invasive bladder cancer (Ta, T1, and Tis; carcinoma in situ) 132 (52%) and muscle invasive bladder cancer (T2–T4, N+, M+) 70 (32.9%). Accurate stage and grade assessment could not be performed in 11 patients (Table 1), i.e., stage was not reported in the medical records and limited pathologic specimen inhibited pathologists from grading.

### 3.2. Bladder Cancer Associated Diagnostic Signature Immunohistochemical Results

Figure 1 depicts stromal expression of the 10 targets of the bladder cancer -associated diagnostic panel in a high-grade non-muscle invasive tumor. The association between the immunophenotype and each target as it relates to disease status is summarized in Table 2. The expression of seven of the 10 bladder cancer -associated diagnostic signature (MMP9, MMP10, PAI1, CA9, APOE, SDC1, and ANG) showed a positive association with bladder cancer diagnosis, while IL8 and A1AT showed a negative correlation with cancer diagnosis. In particular, we found cancer cases expression levels in the third and fourth quartiles for MMP9 (65.4% vs. 12.2% of control), MMP10 (64% vs. 16.3% of control), PAI1 (65% vs. 12.5% of control), CA9 (63.8% vs. 11.5% of control), APOE (61.1% vs. 22.6% of control), SDC1 (64.3% vs. 12.9% of control) and ANG (68.9% vs. 0% of control) to be significantly increased compared to control. Age and race did not correlate with the expression levels of these 10 bladder cancer -associated diagnostic signature (data not shown).

The association between immunophenotype for each of the 10 targets and tumor grade is summarized in Table 3. We found expression of MMP10 in high-grade disease compared to low-grade disease (i.e., 66.7% vs. 60.4%, respectively *p* = 0.016). The association between immunophenotype for each of the 10 targets and tumor stage is noted in Table 4. High stage disease was associated with increased expression (i.e., more third and fourth quartile immunostaining) for MMP10 (60.5% Ta vs. 69.1% T2 vs. 77.7% >T2, *p* = 0.010), PAI1 (59.6% Ta vs. 71.5% T2 vs. 77.2% >T2, *p* = 0.013) and ANG (58.7% Ta vs. 75.9% T2 vs. 77.8% >T2, *p* = 0.003). With CA9, high stage disease was associated with reduced expression (i.e., more first and second quartile immunostaining) 42.2% Ta vs. 37.9% T2 vs. 47.4% >T2, *p* = 0.019. Immunoexpression of SDC1 was noted to be inversely association with tumor stage (80.9% Ta vs. 58.6% T2 vs. 40% >T2, *p* < 0.0001). This is similar to what our group previously reported [20].

### 3.3. CD3^+^ and CD68^+^ Immunohistochemical Results

Figure 2 shows representative expression status for CD3^+^ T cells and CD68^+^ histiocytes in the stroma of a high-grade non-muscle invasive tumor. The relationship between immunophenotype for CD3^+^ and CD68^+^ and tumor grade is summarized in Table 3. Neither CD3^+^ nor CD68^+^ were associated with tumor grade. The relationship between CD68^+^, not CD3^+^, was associated with higher tumor stage (Table 4). High stage disease correlated with increased expression level (i.e., more third and fourth quartile immunostaining) for CD68^+^ (30% Ta vs. 69.5% T2 vs. 61.7% > T2).

### 3.4. Immunophenotype and Survival

The follow-up period for the cohort ranged from 1 to 82 months (median six months), and the mean survival time was 16 months. Univariate analysis indicated that only high stage disease predicted worse overall survival. Similarly, with multivariate analysis, tumor stage < T2 (HR 0.093, 95% CI 0.036–0.235, *p* < 0.0001) independently predicted a worse overall survival (Table 5). Using the Kaplan-Meier survival analysis with the log-rank test, we found significantly reduced overall survival in subjects whose tumors expressed ≥8 of the targets in the diagnostic signature vs. <8 targets in the diagnostic signature (*p* = 0.0271) (Figure 3).

## 4. Discussion

Briefly, our bladder cancer -associated diagnostic signature was derived from voided urine samples. The urine was directly subjected to molecular profiling, proteomic [29,30] and genomic [31,32] for biomarker discovery. This strategy was chosen to avoid the potential drop-out of biomarkers that can occur when translating from tissue-based discovery studies [33,34,35] to biological fluids for assay development. A number of tissue-based biomarkers have translated to urinalysis [36,37,38], but translation can be affected by secretion rate, enzymatic breakdown, or the stability of the protein in the dilute media. Next, four large and unique independent cohorts (totaling 409 bladder cancer subjects and 880 controls) were used to confirm the signature [11,12,13,14]. Then, this diagnostic signature was incorporated into an electrochemiluminescence multiplex assay and validated in three large cohorts [39,40]. The bladder cancer associated diagnostic signature achieved 85% sensitivity, 81% specificity, 93% PPV, and 63% NPV (AUROC 0.8925) for non-invasive detection of bladder cancer [39]. Lastly, pooled data from 1173 patients were analyzed. The log OR for each biomarker was improved by 1.5 or greater with smaller 95% CI in a meta-analysis of the overall cohort compared with each analysis of an individual cohort. The combination of the ten biomarkers showed a higher log OR (log OR: 3.46, 95% CI: 2.60–4.31) than did any single biomarker irrespective of histological grade or disease stage of tumors [41]. Thus, our results justify further advancement of this innovative bladder cancer associated diagnostic signature. In this study, we expanded on our previous work in which we not only confirmed the presence of this urine-based diagnostic signature in epithelial of human bladder tumors. However, we also noted poor prognostic significance when human bladder tumors expressed both IL8 and A1AT.

Maintenance of both normal epithelium and neoplasm is supported by the host’s stroma. The stroma mainly consists of the basement membrane, fibroblasts, extracellular matrix, immune cells, and vasculature. Although most host cells in the stroma possess certain tumor-suppressing abilities, the stroma changes during tumorigenesis. In fact, there is significant crosstalk between tumor epithelial and tumor stroma leading to tumor growth, invasion, and metastasis [42]. A better understanding of the host’s stromal contribution to cancer progression will increase our knowledge about the growth-promoting signaling pathways and hopefully lead to novel therapeutic interventions targeting the tumor stroma. Of the 10 biomarkers in the diagnostic signature, seven biomarkers (MMP9, MMP10, PAI1, CA9, APOE, SDC1 and ANG) were overexpressed in bladder tumor tissue compared to control. Interestingly, IL8 and A1AT were overexpressed in control tissue compared to tumor. Many of the control tissues were associated with benign inflammatory conditions of the bladder (e.g., chronic cystitis, cystitis cystica and cystitis); thus, stromal elevation of IL8 is not surprising. Similar to the results in tumor epithelia, VEGFA expression was not associated with tumor [28]. Additionally, we found that elevated levels of ≥8 of the biomarkers were noted to be associated with a reduced 5-year overall survival (51%) compared to five-year overall survival of 89% when <8 biomarkers were present.

A number of molecular changes have been associated with development and progression of bladder cancer. Such molecular changes include: (1) upregulation of luminal-infiltrate genes, (2) upregulation of basal-like genes, (3) upregulation of luminal-like genes, (4) alterations in expression and regulation of the receptor tyrosine kinases, fibroblast growth factor receptor 3, and members of the epidermal growth factor receptor family, (5) functional down-regulation of the tumor suppressors, p53, pRb, and p16 through deletion, mutation and/or silencing, and (6) upregulation of signaling through RAS and phosphatidylinositol 3-kinase/AKT pathways [9,43,44,45]. Consequently, better understanding the molecular mechanisms associated with initiation, promotion, and progression of bladder cancer is pivotal to effectively prevent, diagnose, and treat bladder cancer.

Since many of the biomarkers in the bladder cancer-associated diagnostic panel are known immunomodulators, we assessed the bladder tumors for the presence of CD3^+^ cells (pan-T cell) and CD68^+^ cells (macrophage) as a snapshot of the tumor immune landscape. High CD3^+^ cells were found to be a good prognostic factor in both NMIBC and MIBC. We could not confirm the results of Sjodahl et al. [46], who demonstrated that increased infiltration of CD3^+^ was associated with improved outcomes in NMIBC and MIBC. Furthermore, we noted CD68^+^ cells to be associated with higher stage disease. This is in agreement with the study by Sjodahl [46]. Unlike the study by Sjodahl et al., we could not confirm that a high ratio between CD68^+^ and CD3^+^ was associated with poor overall survival. We noted macrophage infiltration to be associated with tumor stage, suggestive of its importance in bladder cancer progression and metastasis. To confirm the importance of infiltrating T cells and macrophages, future studies should address the subtypes of T cells and macrophages present in the microenvironment of the bladder tumor since recent studies suggest that during carcinogenesis. For example, CD8^+^ T cells that infiltrate lung tumors were reported to be dysfunctional due to microenvironmental factors, which led to reduced numbers of effector CD8^+^ T cells [47]. Tumor macrophages may polarize to M1 (anti-tumorigenic) or M2 (contributing to carcinogenesis), thus demonstrating different effects on the tumor [48].

## 5. Conclusions

Bladder cancer management is hampered by lack of diagnostic or prognostic markers capable of (a) predicting treatment response and (b) predicting the likely disease course. The molecular characterization of bladder cancer by our diagnostic signature may risk stratify individuals with bladder cancer who have a poor prognosis. The prognostic value of our signature must be independently validated.

## Figures and Tables

**Figure 1 diagnostics-10-00014-f001:**
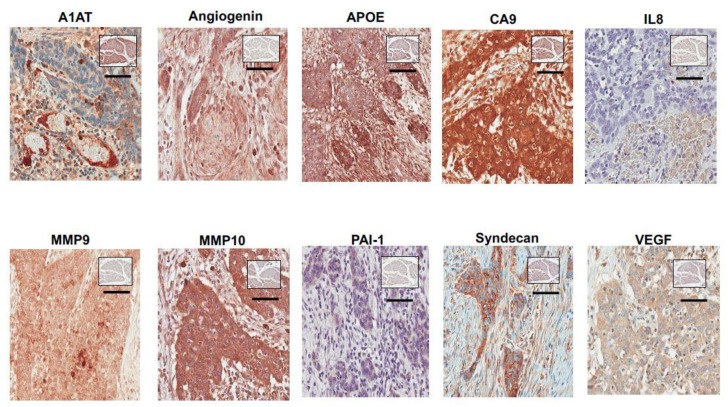
Representative expression status for ANG, CA9, MMP9, MMP10, A1AT, APOE, SDC1, VEGFA, PAI1 and IL8 in high-grade non-muscle invasive bladder tumor. Representative expression status for ANG, CA9, MMP9, MMP10, A1AT, APOE, SDC1, VEGFA, PAI1 and IL8 levels in benign tissue noted in square insert image. All images were captured at 400× magnification.

**Figure 2 diagnostics-10-00014-f002:**
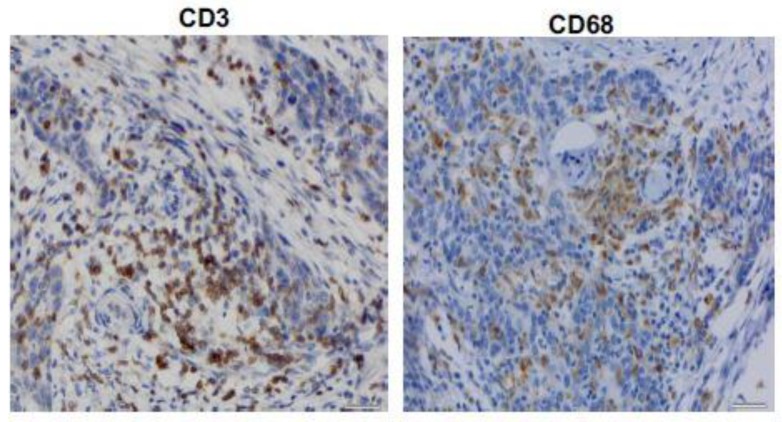
Representative expression status for CD3^+^ and CD68^+^ in high-grade non-muscle invasive bladder tumor. All images were captured at 400× magnification.

**Figure 3 diagnostics-10-00014-f003:**
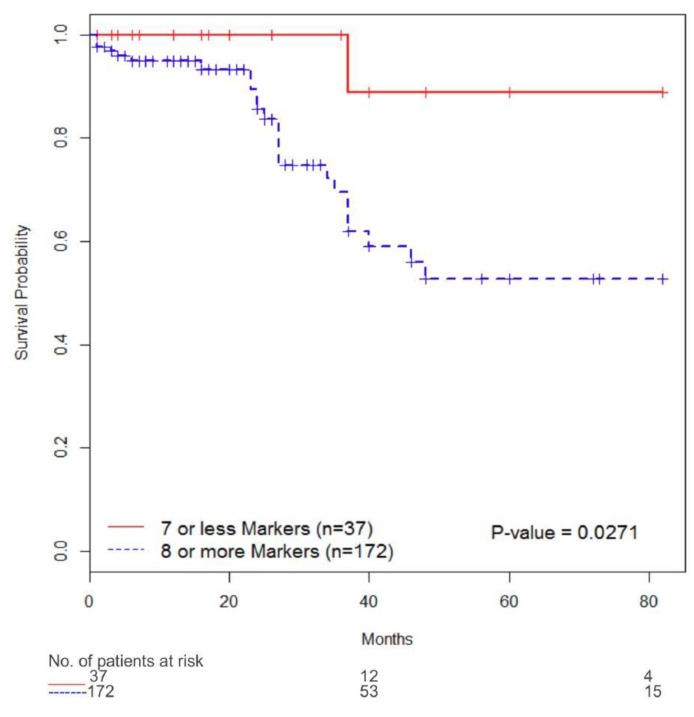
Overall survival analysis of 213 patients with bladder cancer. Overall survival according to immunostaining status of ≥8 targets from diagnostic signature and <8 targets from diagnostic signature.

**Table 1 diagnostics-10-00014-t001:** Demographic, clinical, and pathologic characteristics of the 287 subjects comprising the study cohort.

Features	Bladder Cancer (%) *n* = 213	Controls (%) *n* = 74	*p*-Value
Age (years)			
<65	50 (23.9%)	74 (100.0%)	<0.0001
≥65	150 (70.4%)	0 (0.0%)	
Unavailable	12 (5.6%)		
Sex			
Female	48 (22.5%)	16 (21.6%)	0.154
Male	153 (71.8%)	58 (78.4%)	
Unavailable	12 (5.6%)		
Race			
Caucasian	162 (76.5%)	61 (82.4%)	0.672
Other	30 (14.1%)	7 (9.5%)	
Unavailable	20 (9.4%)	6 (8.1%)	
Tumor grade			
High-grade	176 (82.6%)		
Low-grade	26 (12.2%)		
Unavailable	11 (5.2%)		
Tumor stage			
Ta	50 (23.5%)		
Tis	20 (9.4%)		
T1	62 (29.1%)		
T2	31 (14.6%)		
≥T3, N+ or M+	39 (18.2%)		
Unavailable	11 (5.2%)		
Recurrence			
Yes	70 (32.9%)		
No	143 (67.1%)		

**Table 2 diagnostics-10-00014-t002:** Relationship between immunochemical features and disease status.

Target Expression	Bladder Cancer, %	Benign, %	*p*-Value	Target Expression	Bladder Cancer, %	Benign, %	*p*-Value
VEGFA				APOE			
1	35 (21.3%)	24 (32.4%)	0.2361	1	36 (18.7%)	30 (42.3%)	**<0.0001**
2	43 (26.2%)	17 (23.0%)		2	41 (21.2%)	25 (35.2%)	
3	45 (27.4%)	14 (18.9%)		3	56 (29.0%)	10 (14.1%)	
4	41 (25.0%)	19 (25.7%)		4	60 (31.1%)	6 (8.5%)	
MMP9				A1AT			
1	22 (12.1%)	42 (56.8%)	**<0.0001**	1	60 (32.3%)	5 (6.8%)	**<0.0001**
2	41 (22.5%)	23 (31.1%)		2	55 (29.6%)	10 (13.5%)	
3	57 (31.3%)	7 (9.5%)		3	44 (23.7%)	21 (28.4%)	
4	62 (34.1%)	2 (2.7%)		4	27 (14.5%)	38 (51.4%)	
MMP10				SDC1			
1	36 (19.7%)	28 (37.8%)	**<0.0001**	1	22 (11.9%)	41 (58.6%)	**<0.0001**
2	30 (16.4%)	34 (45.9%)		2	44 (23.8%)	20 (28.6%)	
3	55 (30.1%)	9 (12.2%)		3	57 (30.8%)	7 (10.0%)	
4	62 (33.9%)	3 (4.1%)		4	62 (33.5%)	2 (2.9%)	
PAI1				ANG			
1	13 (7.1%)	50 (69.4%)	**<0.0001**	1	5 (2.6%)	61 (84.7%)	**<0.0001**
2	51 (27.9%)	13 (18.1%)		2	55 (28.5%)	11 (15.3%)	
3	60 (32.8%)	4 (5.6%)		3	66 (34.2%)	0 (0%)	
4	59 (32.2%)	5 (6.9%)		4	67 (34.7%)	0 (0%)	
IL8							
1	62 (31.6%)	4 (5.7%)	**<0.0001**				
2	49 (25.0%)	18 (25.7%)					
3	39 (19.9%)	27 (38.6%)					
4	46 (23.5%)	21 (30.0%)					
CA9							
1	23 (11.7%)	43 (61.4%)	**<0.0001**				
2	48 (24.5%)	19 (27.1%)					
3	60 (30.6%)	6 (8.6%)					
4	65 (33.2%)	2 (2.9%)					

Bolded denotes significance.

**Table 3 diagnostics-10-00014-t003:** Relationship between immunochemical features and tumor grade.

Target Expression	Low-Grade, %	High-Grade, %	*p*-Value *	Target Expression	Low-Grade, %	High-Grade, %	*p*-Value
VEGFA				APOE			
1	2 (8.3%)	33 (23.7%)	0.146	1	3 (12.5%)	33 (20.2%)	0.630
2	5 (20.8%)	37 (26.6%)		2	6 (25.0%)	35 (21.5%)	
3	7 (29.2%)	38 (27.3%)		3	9 (37.5%)	44 (27.0%)	
4	10 (41.7%)	31 (22.3%)		4	6 (25.0%)	51 (31.3%)	
MMP9				A1AT			
1	3 (12.5%)	18 (11.8%)	0.673	1	6 (27.3%)	52 (33.1%)	0.257
2	3 (12.5%)	36 (23.5%)		2	10 (45.5%)	43 (27.4%)	
3	8 (33.3%)	48 (31.4%)		3	5 (22.7%)	36 (22.9%)	
4	10 (41.7%)	51 (33.3%)		4	1 (4.5%)	26 (16.6%)	
MMP10				SDC1			
1	8 (33.3%)	27 (17.6%)	**0.016**	1	1 (4.0%)	21 (13.2%)	0.135
2	0 (0%)	29 (19.0%)		2	3 (12.0%)	40 (25.2%)	
3	10 (41.7%)	43 (28.1%)		3	8 (32.0%)	49 (30.8%)	
4	6 (25.0%)	54 (35.3%)		4	13 (52.0%)	49 (30.8%)	
PAI1				ANG			
1	3 (12.5%)	8 (5.3%)	0.292	1	1 (4.2%)	4 (2.5%)	0.248
2	5 (20.8%)	46 (30.3%)		2	6 (25.0%)	48 (29.4%)	
3	10 (41.7%)	47 (30.9%)		3	5 (20.8%)	58 (35.6%)	
4	6 (25.0%)	51 (33.6%)		4	12 (50.0%)	53 (32.5%)	
IL8				CD3^+^			
1	8 (33.3%)	52 (31.7%)	0.592	1	6 (28.6%)	32 (25.0%)	0.904
2	5 (20.8%)	40 (24.4%)		2	5 (23.8%)	32 (25.0%)	
3	3 (12.5%)	35 (21.3%)		3	4 (19.0%)	33 (25.8%)	
4	8 (33.3%)	37 (22.6%)		4	6 (28.6%)	31 (24.2%)	
CA9				CD68^+^			
1	3 (13.0%)	20 (12.0%)	0.464	1	5 (22.7%)	34 (25.4%)	0.482
2	5 (21.7%)	40 (24.1%)		2	6 (27.3%)	33 (24.6%)	
3	10 (43.5%)	48 (28.9%)		3	8 (36.4%)	32 (23.9%)	
4	5 (21.7%)	58 (34.9%)		4	3 (13.6%)	35 (26.1%)	

Bolded denotes significance.

**Table 4 diagnostics-10-00014-t004:** Relationship between immunochemical features and tumor stage.

Target Expression	Ta	Tis	T1	T2	≥T3, N+ or M+	*p*-Value
VEGFA						
1	7 (17.5%)	4 (23.5%)	11 (20.4%)	8 (33.3%)	5 (17.9%)	0.854
2	10 (25.0%)	6 (35.3%)	12 (22.2%)	7 (29.2%)	7 (25.0%)	
3	11 (27.5%)	4 (23.5%)	17 (31.5%)	3 (12.5%)	10 (35.7%)	
4	12 (30.0%)	3 (17.6%)	14 (25.9%)	6 (25.0%)	6 (21.4%)	
MMP9						
1	5 (11.6%)	1 (7.1%)	5 (8.9%)	7 (24.1%)	3 (8.6%)	0.1389
2	6 (14.0%)	7 (50.0%)	16 (28.6%)	3 (10.3%)	7 (20.0%)	
3	17 (39.5%)	3 (21.4%)	16 (28.6%)	7 (24.1%)	13 (37.1%)	
4	15 (34.9%)	3 (21.4%)	19 (33.9%)	12 (41.4%)	12 (34.3%)	
MMP10						
1	14 (32.6%)	5 (35.7%)	12 (21.8%)	3 (10.3%)	1 (2.8%)	**0.0103**
2	3 (7.0%)	4 (28.6%)	9 (16.4%)	6 (20.7%)	7 (19.4%)	
3	15 (34.9%)	4 (28.6%)	17 (30.9%)	5 (17.2%)	12 (33.3%)	
4	11 (25.6%)	1 (7.1%)	17 (30.9%)	15 (51.7%)	16 (44.4%)	
PAI1						
1	6 (14.3%)	2 (13.3%)	1 (1.8%)	2 (7.1%)	0 (0%)	**0.013**
2	11 (26.2%)	8 (53.3%)	18 (32.1%)	6 (21.4%)	8 (22.9%)	
3	13 (31.0%)	5 (33.3%)	14 (25.0%)	8 (28.6%)	17 (48.6%)	
4	12 (28.6%)	0 (0%)	23 (41.1%)	12 (42.9%)	10 (28.6%)	
IL8						
1	18 (39.1%)	7 (43.8%)	17 (28.3%)	5 (17.2%)	13 (35.1%)	0.7881
2	8 (17.4%)	3 (18.8%)	17 (28.3%)	8 (27.6%)	9 (24.3%)	
3	9 (19.6%)	4 (25.0%)	12 (20.0%)	6 (20.7%)	7 (18.9%)	
4	11 (23.9%)	2 (12.5%)	14 (23.3%)	10 (34.5%)	8 (21.6%)	
CA9						
1	3 (6.7%)	0 (0%)	6 (10.0%)	8 (27.6%)	6 (15.8%)	**0.0192**
2	16 (35.6%)	5 (29.4%)	9 (15.0%)	3 (10.3%)	12 (31.6%)	
3	13 (28.9%)	7 (41.2%)	18 (30.0%)	7 (24.1%)	13 (34.2%)	
4	13 (28.9%)	5 (29.4%)	27 (45.0%)	11 (37.9%)	7 (18.4%)	
APOE						
1	9 (19.1%)	2 (12.5%)	14 (23.7%)	6 (20.7%)	5 (13.9%)	0.967
2	12 (25.5%)	3 (18.8%)	10 (16.9%)	8 (27.6%)	8 (22.2%)	
3	14 (29.8%)	6 (37.5%)	15 (25.4%)	7 (24.1%)	11 (30.6%)	
4	12 (25.5%)	5 (31.3%)	20 (33.9%)	8 (27.6%)	12 (33.3%)	
A1AT						
1	16 (36.4%)	4 (26.7%)	22 (37.9%)	6 (21.4%)	10 (29.4%)	0.403
2	18 (40.9%)	4 (26.7%)	15 (25.9%)	8 (28.6%)	8 (23.5%)	
3	7 (15.9%)	4 (26.7%)	15 (25.9%)	7 (25.0%)	8 (23.5%)	
4	3 (6.8%)	3 (20.0%)	6 (10.3%)	7 (25.0%)	8 (23.5%)	
SDC1						
1	3 (7.1%)	7 (36.8%)	2 (3.4%)	5 (17.2%)	5 (14.3%)	**<0.0001**
2	5 (11.9%)	5 (26.3%)	10 (16.9%)	7 (24.1%)	16 (45.7%)	
3	10 (23.8%)	2 (10.5%)	24 (40.7%)	11 (37.9%)	10 (28.6%)	
4	24 (57.1%)	5 (26.3%)	23 (39.0%)	6 (20.7%)	4 (11.4%)	
ANG						
1	2 (4.3%)	0 (0%)	0 (0%)	3 (10.3%)	0 (0%)	**0.003**
2	17 (37.0%)	11 (64.7%)	14 (23.7%)	4 (13.8%)	8 (22.2%)	
3	13 (28.3%)	1 (5.9%)	22 (37.3%)	10 (34.5%)	17 (47.2%)	
4	14 (30.4%)	5 (29.4%)	23 (39.0%)	12 (41.4%)	11 (30.6%)	
CD3^+^						
1	12 (31.6%)	1 (12.5%)	14 (29.2%)	1 (4.8%)	10 (29.4%)	0.444
2	8 (21.1%)	1 (12.5%)	10 (20.8%)	8 (38.1%)	10 (29.4%)	
3	7 (18.4%)	3 (37.5%)	13 (27.1%)	5 (23.8%)	9 (26.5%)	
4	11 (28.9%)	3 (37.5%)	11 (22.9%)	7 (33.3%)	5 (14.7%)	
CD68^+^						
1	17 (42.5%)	5 (55.6%)	9 (18.0%)	2 (8.7%)	6 (17.7%)	**0.010**
2	11 (27.5%)	1 (11.1%)	15 (30.0%)	5 (21.7%)	7 (20.6%)	
3	9 (22.5%)	3 (33.3%)	13 (26.0%)	7 (30.4%)	8 (23.5%)	
4	3 (7.5%)	0 (0.0%)	13 (26.0%)	9 (39.1%)	13 (38.2%)	

Bolded denotes significance.

**Table 5 diagnostics-10-00014-t005:** Overall Survival.

	Univariate Analyses ^1^						Multivariate Analyses ^1^			
	*n*	Median ^2^	HR ^3^	LCL	UCL	*p* ^4^	HR ^3^	LCL	UCL	*p* ^4^
Age (years)										
≤65	56		0.356	0.107	1.188	0.075	0.376	0.103	1.368	0.138
>65	153									
Sex										
Female	46		1.222	0.513	2.908	0.646				
Male	163									
Tumor Grade										
Low	28		0.489	0.115	2.072	0.314				
High	181									
Tumor Stage										
<T2	144		0.093	0.036	0.235	<0.0001	0.107	0.034	0.337	**0.0001**
≥T2	65	34								
VEGFA										
1–2	79		1.000	0.375	2.670	0.999				
3–4	78									
MMP9										
1–2	84		0.950	0.425	2.125	0.900				
3–4	84									
MMP10										
1–2	85	46	1.224	0.547	2.741	0.616				
3–4	85									
PAI1										
1–2	86		0.708	0.304	1.646	0.416				
3–4	86									
IL8										
1–2	91		0.706	0.320	1.557	0.379				
3–4	90									
CA9										
1–2	91		1.897	0.789	4.563	0.141	1.348	0.536	3.390	
3–4	91									
APOE										
1–2	93		0.803	0.352	1.828	0.595				
3–4	87									
A1AT										
1–2	84		0.475	0.213	1.062	0.060	0.750	0.292	1.926	
3–4	83	37								
SDC1										
1–2	86	48	1.492	0.670	3.325	0.318				
3–4	90									
ANG										
1–2	90		0.969	0.423	2.222	0.940				
3–4	90									
CD3^+^										
1–2	74	37	1.125	0.464	2.729	0.792				
3–4	74									
CD68^+^										
1–2	78		0.524	0.223	1.230	0.126	1.334	0.505	3.524	0.561
3–4	77	37								
CD68^+^/CD3^+^										
CD3^+^ 1–2 & CD68^+^ 1–2	40		0.434	0.112	1.685	0.258				
CD3^+^ 1–2 & CD68^+^ 3–4	33		0.523	0.153	1.792					
CD3^+^ 3–4 & CD68^+^ 1–2	32	35	1.422	0.446	4.534					
CD3^+^ 3–4 & CD68^+^ 3–4	41	48								

^1^ Univariate analyses are unadjusted. Multivariate analyses are adjusted for age, sex, tumor grade, tumor stage, and lymph nodes. ^2^ Median is the median years progression free as estimated from a parametric model. ^3^ HR is the hazard ratio from a semiparametric (Cox proportional hazards) model. The 95% confidence limits (LCL and UCL) are shown. ^4^ The expression p-values are for trend. Bolded denotes significance.

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
