# Peer review of "Prognostic Significance of Lymphocyte Infiltration and a Stromal Immunostaining of a Bladder Cancer Associated Diagnostic Panel in Urothelial Carcinoma"

_diagnostics, 2019, doi:10.3390/diagnostics10010014_

Round 1

Reviewer 1 Report

Furuya et al performed a very interesting study titled “Prognostic significance of lymphocyte infiltration and a stromal immunostaining of a bladder cancer associated diagnostic panel in urothelial carcinoma” in which they analyze the expression of 10 proteins, using IHC in stromal tissue of bladder cancer. The paper is well written and the study logically conducted, despite this some issue need to be addressed before the manuscript reaches the standards of the journal.

Not clear how the authors were sure they were analyzing stromal tissue and not tumor/epithelial tissue. It would benefit the manuscript if you could further detail on this. How were the controls defined? What pathologies did the controls have? Add these data and discuss the implication of this. For example, were the pathologies affecting the controls inflammatory diseases? Are the biomarkers affected by these? Also, update table 1 and add the underling pathologies of the control group. Perform statistical comparison regarding age and gender between cancer and control group. Update table 1. Provide data regarding the race of the control patients and compare the data with that of the cancer patients (Fisher’s test). In Figure 1 images from benign tissues is missing. In figure captions you mention also benign tissues. In table 2 – specify the significance of “*”. Check also if there is any association between race and the level of expressed proteins, similar to the relationship between immunochemical features and disease status. Regarding CD68 and CD3, are these proteins expressed in benign lesions, add data showing the expression level of these two markers in benign/healthy tissue (add the images to figure 2 as controls). Figure 3: add the patients at risk under the x axis.

Author Response

Reviewer #1

Not clear how the authors were sure they were analyzing stromal tissue and not tumor/epithelial tissue. It would benefit the manuscript if you could further detail on this.

The slides were scanned using the Aperio Scanscope Cs (Aperio Technologies, Vista, CA) as high-resolution images (20x objective). Images were visualized using Image Scope (Aperio, Vista, CA, USA). The immunoreactivity staining patterns within the stroma were noted. A prescribed algorithm developed by Aperio was used to assess staining intensity of the tissue and percent of cells/tissue staining for each target within the stromal only (Note – We have previously used an algorithm to report the staining within the epithelial component of the tumor (28)).

How were the controls defined? What pathologies did the controls have? Add these data and discuss the implication of this. For example, were the pathologies affecting the controls inflammatory diseases? Are the biomarkers affected by these? Also, update table 1 and add the underling pathologies of the control group.

47 autopsies for non-BCa cause of death and 27 bladder neck specimens from radical prostatectomy specimens. None of the controls had diagnosed inflammatory conditions. 

Perform statistical comparison regarding age and gender between cancer and control group. Update table 1. Provide data regarding the race of the control patients and compare the data with that of the cancer patients (Fisher’s test).

Great point.  This was added.

In Figure 1 images from benign tissues is missing. In figure captions you mention also benign tissues.

Benign section is inserted in each panel.  This insert now is outlined to better highlight this.

In table 2 – specify the significance of “*”.

This was corrected.

Check also if there is any association between race and the level of expressed proteins, similar to the relationship between immunochemical features and disease status.

Interesting point.  We performed this analysis and did not see any relationships between race (and age checked too) in expression of these proteins. 

Regarding CD68 and CD3, are these proteins expressed in benign lesions, add data showing the expression level of these two markers in benign/healthy tissue (add the images to figure 2 as controls).

CD68 (macrophage) and CD3 (T-cells) are rarely seen in benign/healthy tissue. 

Figure 3: add the patients at risk under the x axis.

This was added. 

Reviewer 2 Report

The authors build upon their previous study focusing on the diagnostic power of epithelial expression markers for bladder cancer. In this study, the authors also look at stromal and immune cell markers to expand their diagnostic capability. The most striking observation they make is that the differentially expressed markers in the stromal compartment of bladder cancer also predicts patient prognosis. The study compares IHC staining from tumor samples to benign controls. I would strongly sugges the authors to address the following points:

Line 133: MMP10 expression although statistically significant, seems to have a very small effect size (66% vs 60%). Does this not suggest that MMP10 expression would likely be a false positive in a majority of benign samples?

In all Tables: It would be easier to read through the tables if the statistically signifcant prognostic markers are in boldface or highlighted in some way.

Line 145: The authors made a claim to look at CD3 and CD68 as potential biomarkers, however, these were found to be largely unassociated with disease progression. It might be pertinent for the authors to why they initially included these in the panel, and a discussion on why they don't predict survival.

Author Response

Line 133: MMP10 expression although statistically significant, seems to have a very small effect size (66% vs 60%). Does this not suggest that MMP10 expression would likely be a false positive in a majority of benign samples?

This reported difference is what seen between high-grade (66%) and low-grade disease (60%).  When it comes to cancer and controls the difference is 64% vs 16%.  The power in our signature rests in its combinatorial effect which protects it from error, i.e., false positive or false negatives. 

In all Tables: It would be easier to read through the tables if the statistically signifcant prognostic markers are in boldface or highlighted in some way.

This has been added. 

Line 145: The authors made a claim to look at CD3 and CD68 as potential biomarkers, however, these were found to be largely unassociated with disease progression. It might be pertinent for the authors to why they initially included these in the panel, and a discussion on why they don't predict survival.

Yes we hypothesized that our BCa signature could correlate with CD3 and CD68 tissue levels.  Our hypothesis was not confirmed.  This was clarified in the text. 

Round 2

Reviewer 1 Report

The authors addressed all my concerns and the manuscript reaches the standards of the journal. 

Moreover, the authors should prepare a new figure 3, on the x axis should be the Months and under the months should be number of patients at risk. In its current form the figure is hard to understand. 

Author Response

Figure modified.